# An Eight-Year Survey on Aflatoxin B1 Indicates High Feed Safety in Animal Feed and Forages in Northern Italy

**DOI:** 10.3390/toxins14110763

**Published:** 2022-11-04

**Authors:** Luca Ferrari, Francesca Fumagalli, Nicoletta Rizzi, Elisa Grandi, Serena Vailati, Michele Manoni, Matteo Ottoboni, Federica Cheli, Luciano Pinotti

**Affiliations:** 1Department of Veterinary Medicine and Animal Sciences (DIVAS), Università degli Studi di Milano, Via dell’ Università, 26900 Lodi, Italy; 2Associazione Regionale Allevatori della Lombardia (ARAL), Via Kennedy, 26013 Crema, Italy; 3CRC I-WE (Coordinating Research Centre: Innovation for Well-Being and Environment), Università degli Studi di Milano, 20133 Milan, Italy

**Keywords:** aflatoxins, AFB1, mycotoxins, animal feed, maize, monitoring program

## Abstract

Aflatoxins (AFs) remain the main concern for the agricultural and dairy industries due to their effects on the performances and quality of livestock production. Aflatoxins are always unavoidable and should be monitored. The objective of this paper is to bring to light a significant volume of data on AF contamination in several animal feed ingredients in Northern Italy. The Regional Breeders Association of Lombardy has been conducting a survey program to monitor mycotoxin contamination in animal feeds, and in this paper, we present data relating to AFB1 contamination. In most cases (95%), the concentrations were low enough to ensure compliance with the European Union’s (EU’s) maximum admitted levels for animal feed ingredients. However, the data show a high variability in AF contamination between different matrices and, within the same matrix, a high variability year over year. High levels of AFs were detected in maize and cotton, especially in the central part of the second decade of this century, i.e., 2015–2018, which has shown a higher risk of AF contamination in feed materials in Northern Italy. Variability due to climate change and the international commodity market affect future prospects to predict the presence of AFs. Supplier monitoring and control and reduced buying of contaminated raw materials, as well as performing analyses of each batch, help reduce AF spread.

## 1. Introduction

The feed supply chain is crucial for livestock systems, feed supply, and feed/food safety. Feed origin, processing, handling, and storage can affect both quality and safety at different levels [1]. Mycotoxins are one of the most important safety risks in the feed industry and supply chain security [2,3]. 

Aflatoxins (AFs) are a group of natural fungal toxins found in a wide range of animal feeds and human foods. Aflatoxins are mainly produced by *Aspergillus flavus*, *Aspergillus parasiticus*, and rarely by *Aspergillus nominus* [4,5]. Among the AFs, aflatoxin B1 (AFB1), aflatoxin B2 (AFB2), aflatoxin G1 (AFG1), aflatoxin G2 (AFG2), and aflatoxin M1 (AFM1) contamination in feed and food poses serious risks to animal and human health. Based on scientific evidence, AFB1 and AFG1 are classified by the International Agency for Research on Cancer (IARC) in Group 1 [6], along with certain human carcinogens and AFM1 in group 2B, as potentially carcinogenic to humans [7]. AFB1 remains the most dangerous and occurring mycotoxin, and related hazards concern not only chronic but also acute exposure. Although the mechanism of action is not yet well known, several authors report that AFB1 toxicity is due to an interaction with the macromolecules of biological organisms, such as proteins, phospholipids, and nucleic acids, with the consequent formation of different adducts, which interfere with the normal physiological functioning [8,9,10,11]. Similarly, chronic exposure to AFs has serious consequences and has been linked to hepatocellular carcinoma [12], stunted growth [13], and impaired immunity [14] in both humans and animals. 

Livestock are also at risk for the serious consequences of chronic exposure to AFs. Poultry is generally very sensitive to AFB1, and the hazards are associated with low productivity, i.e., reduced feed intake with a reduction in body weight or small egg production [15,16,17] and high disease susceptibility, which can have negative impacts on the poultry producers’ income, as well as affect human health [18]. Pigs are considered relatively susceptible to AFB1, with symptoms such as reduced average daily weight gain and reduction in overall growth rate [19,20,21]. Ruminants are generally more resistant to mycotoxin toxic effects than monogastric animals [2]. In fact, rumen microbiota can degrade or inactivate many toxic molecules, such as zearalenone (ZEA) (90%) and the trichothecene T-2 toxin (completely) but had no effects on AFB1. This is due to the inhibitory action of AFs on ruminal bacteria growth [22]. The main problem for ruminants is the carry-over of AFB1 to AFM1 to milk and dairy products [23]. 

AFB1 contaminates several feed/food crops, such as maize, barley, rice, and crude vegetable oils [10,11], and its presence cannot be predicted and completely avoided, especially in tropical countries where the humidity rate is always particularly high [2]. Although crops from tropical and/or sub-tropical areas are affected more frequently and severely by AF contamination, temperate areas, such as Europe, could be of increasing importance due to climate change, enhancing risks for human and animal health. Climate change, in particular, has been reported as a driver for emerging food and feed safety issues worldwide, and its expected impact on the presence of AFs in food and feed is of great concern. AFB1 is predicted to become a food safety issue in maize in Europe, especially in a warming scenario (+2 °C scenario), foreseen for the next years [1,2]. Given the potential health repercussions, more than 100 countries around the world have established mycotoxin regulations [3]. In Europe, AFB1 in feed is regulated by the Commission Regulation (EC) No 1881/2006, while the Food and Drug Administration (FDA) has established recommended action levels for the United States of America [2]. However, differences in acceptable AF levels between countries and regions can lead to distortions in international commodity markets, with more contaminated crops excluded from countries with stricter regimes and, potentially, traded more extensively in countries with less regulatory controls [24,25]. It is important to emphasise that food and feed commodities are always contaminated, not only by Afs but by mycotoxins’ co-occurrence, which represents the rule and not the exception, as confirmed by a European multi-mycotoxin contamination study which indicated that 75% to 100% of animal feed samples contain more than one mycotoxin [26]. 

Among the crops susceptible to contamination by AFs, maize is one of the most important feed resources for the livestock sector [27,28]. Maize is widely grown in Northern Italy; in 2003, significant problems occurred due to AF contamination of this crop. The summer was particularly dry and hot, with maize crops stressed by drought and lack of water, and the consequence was high AFB1 contamination. Important repercussions have occurred in the dairy industry with high levels of AFM1 in milk. Based on this evidence, in March 2016, the regional authorities in Lombardy activated the "Extraordinary operating procedures for the prevention and management of the risk of contamination by AFs in the dairy supply chain and in the production of maize" [29]. These aspects have been very well addressed by Roila et al. (2021), who studied the occurrence of AFM1 in cow and ewe milk over a seven-year period (2014–2020) in the central part of Italy. The average concentration of AFM1 in cow’s milk ranged from 9 to 15 ng/kg, while in ewe’s milk, the average concentration ranged from 9 to 13 ng/kg [30]. The average amount of AFM1 exposure ranged from 0.05 to 1.95 mg/kg bw/day, with the main contributor being represented by drinking milk, followed by the consumption of soft cheeses [30]. Such scenarios evidence the necessity for implementing AF monitoring, starting from the feed.

Despite efforts to control fungal contamination, mycotoxins are ubiquitous in nature and are regularly produced in crops susceptible to contamination around the world. Although wide mycotoxin contamination surveys have been carried out worldwide [26,31,32], there are very few monitoring studies on AFs in feed in Italy, many of which are conducted on a limited number of samples and very limited in time [33,34,35]. In this context, the aim of this study was to evaluate AFB1 contamination in raw materials and feed for animal nutrition in Northern Italy. The survey was conducted by the Regional Breeders Association of Lombardy, referring to the years 2013−2020. 

## 2. Results

The results showing the levels of AFB1 in the different samples collected in Italy are reported in Table 1. In particular, only 597 samples of the 10,280 samples analysed in the period from 2013–2020 (corresponding to 5.7% of the total samples) exceed the level of 20 µg/kg allowed by the European Union for animal feed ingredients, while all the other samples (corresponding to 94.3%) comply with the European legal limits.

Correlating the type of the matrix and the level of contamination, maize and cotton were the matrices with the greater number of positive samples and highest contamination values of 1074 µg/kg and 728 µg/kg, respectively.

Given these results, the analysis of the data was focused on the maize and cotton that showed the highest susceptibility to AFB1 contamination. Figure 1 shows the mean values of AFB1 contamination found over the years for cotton and different types of maize matrices. 

When maize was considered, it showed the highest concentration of AFB1 during 2015 and 2016, when AFB1 contamination reached the mean values of 15.7 and 12.4 µg/kg, respectively (Figure 1A). In the case of silages such as maize silage and high moisture maize (Figure 1B), the mean values of AFB1 contamination were almost constant over the years, with the only exception being those of 2015 for high-moisture maize. This scenario is also confirmed in Figure 2, where the percentage of positive samples is reported.

For cotton, the trend of AFB1 contamination over the years showed a peak of contamination in 2015 and 2018 (Figure 1A and Figure 2), while more recently (i.e., 2020), the contamination levels observed were lower (Figure 1A). 

Combining these results, it can be speculated that the central part of the second decade of this century, i.e., 2015–2018, has shown a higher risk of AF contamination in feed materials in Northern Italy, although monitoring control programs, such as the Regional Breeders Association of Lombardy, help to maintain the contamination under control. 

## 3. Discussion

### 3.1. AFB1 Occurrence in Feed in Northern Italy

The monitoring of AF contamination in animal feed is a critical point as mycotoxins in animal feed can exert toxic effects on animals and be transferred into products of animal origin. Mycotoxins are also naturally occurring metabolites of fungi and molds whose presence in feedingstuffs is mostly unavoidable [36]. For this reason, since 2004, the Regional Breeders Association of Lombardy has been conducting a survey program to monitor mycotoxin contamination in animal feed. In this study, the occurrence of AFB1 was analysed in 10,280 feed samples during an eight-year period (September 2013 to July 2021). The results, based on mean values of AFB1 contamination, indicate that almost the totality of the matrices analysed in this study comply with the legal limits set by the European Union for AFB1 in animal feed ingredients. In detail, the results show that about 95% of the analysed samples have an incidence of contamination below the legal limit, set by the European Union, of 20 µg/kg for feed materials. In particular, the majority of these samples (about 84%) had an incidence of contamination below 5µg/kg of AFB1. Although the average AFB1 contamination of the analysed samples comply with European regulations, 5.7% (597 samples) are positive. These samples do not comply with the legal limits set by the European Union, and, for this reason, they cannot reach the market with significant economic losses for producers. 

When the contaminated samples are considered, our results indicate that maize and cotton are the most contaminated samples, confirming that they are the crops most susceptible to AF contamination, as reported in several worldwide surveys [22,26,37,38], unlike rice or wheat, which show little susceptibility to the attack of fungi and molds [39]. The reasons for that are several environmental factors, such as physical factors (moisture and temperature) and biological factors (fungal species, weeds, and insect injuries), as well as nutritional factors, such as carbon, amino acids, nitrogen, and lipid content [40]. 

### 3.2. AFB1 Occurrence in Feed in Northern Italy: Year-to-Year Variation

The results of our study indicate that cotton and maize present a significant year-to-year variation in AFB1 occurrence.

#### 3.2.1. Cotton 

Looking at the average contamination over the years (Figure 1A), there was a consistent increase in cotton contamination in two distinct years, 2015 and 2018. In fact, while the average contamination in other years was about 4–5 µg/kg, the average of AFB1 rose to 14.1 µg/kg in 2015 and to 18.8 µg/kg in 2018. This increase in contamination was only partially reflected in the maximum level of AFB1, so much so that in 2018 there was a maximum level of 375 µg/kg against the values in the range of 60–288 µg/kg in the other years. In 2015, there was also an increase in the maximum level of contamination, with a value of 728.8 µg/kg. 

Cotton is an industrial crop, grown mainly for the textile, food and cosmetics sectors. Cotton seeds are the seeds of the cotton plant. Cotton seeds are ovoid, 3.5–10 mm long. Commercially available cotton seeds are usually the by-product of the production of cotton fibre by a cotton gin, which separates the lint from the seeds [41]. Raw cotton seeds and cotton seeds cake can be a good source of protein for animals, especially for ruminants in general and dairy cattle in particluar, since cotton seeds have some potential in milk production. However, their use is limited due to their heavy AF contamination. 

Global cotton production was equal to 25 million tons per year in 2020; the producing countries were China, India, the USA, Pakistan, Uzbekistan, Brazil, Turkey, Australia, Greece, and Turkmenistan. The main exporters were the USA, India, Brazil, Australia, Burkina Faso, Greece, Uzbekistan, the Ivory Coast, and Turkmenistan. 

The heavy contamination of raw cotton seeds with AFB1 has to be avoided, due to its toxic effects on animals after the ingestion of contaminated cotton seeds, as well as the heavy contamination of milk with AFM1. Global studies of AF occurrence in cotton are rare, but according to a study in Texas and one in Pakistan, AFs are higher due to specific weather conditions [42,43]. In fact, AF contamination depends not only on the type of crop but also on the climatic conditions in which the crop grows [28]. In particular, *A. flavus*, one of the main producers responsible for AFB1 production, can grow on the mature crop when it is exposed to high humidity and warm temperature either before or after harvest [39]. Tropical and subtropical climates, where a hot and humid environment is common, characterizes the main producing and exporting countries, such as India, Pakistan, and Greece, enhancing the risk of heavy AFB1 contamination. Under favorable tropical and subtropical conditions (high temperatures and high humidity), these molds can invade food/feed crops [36], even if temperate areas, such as Europe, could be of increasing importance due to climate change.

Since the trend of AFB1 contamination can change over the years and from one country to another, monitoring control programs, such as the one presented in our study, help to maintain the contamination under control. 

#### 3.2.2. Maize, Maize Silage, and High-Moisture Maize

In Italy, maize is widely grown in the northern regions, 602,900 hectares of which approximately 31,400 hectares are cultivated in Lombardy [44]. Ensiling is a widespread practice to preserve fodder [45], and in the Po Valley, it is mostly used to produce maize silage for dairy cows. A large part of maize produced in Lombardy is intended for silage production, and this, in terms of mycotoxin control, can be positive, given the data of this study, of which maize silage and high-moisture maize were shown to be less susceptible to the AF contamination, as reported in several other studies [46,47]. In particular, in our study, maize showed the highest percentage of positive samples year-over-year (expressed as the samples with contamination beyond the limit of 20 µg/kg AFB1 set by the EU), with a percentage in many years over 4% of the positive samples (Figure 2). These values are in contrast to the values reported by Šegvić Klarić et al. (2009), where the authors reported 33% of contaminated samples, all in the range between 2.7 and 4.5 µg/kg [48]. However, the data we have reported here have to be considered with caution since, on average, the level of contamination was 9.1 µg/kg of AFB1 (i.e., quite low).

For 2013 and 2014, the maximum levels of AFB1 detected were 196 and 275 µg/kg, respectively, but the level of contamination increased significantly in 2015 and 2016 when contamination peaked at 1074 and 1003 µg/kg, respectively. The increase in maize contamination could be seen to a lesser extent also by looking at the percentage of samples that exceeded the limit of 20 µg/kg set by EU legislation (Figure 2). In 2015, there was a significant increase of approximately 10.15% compared to relatively low contamination in 2013 and 2014, when the percentage of samples over the limit was 7.46 and 3.26%, respectively. 

Analysing the data relating to maize, it is important to underline how the contamination trend was greatly influenced by the nature of the matrix. Taking, for example, the average level of AFs, the samples of grain and flour consistently showed higher values between high-moisture maize and silage maize. The average for flour and grain was around 3–4 µg/kg, while for high-moisture maize and silage maize, the average was about 0.9–1 µg/kg and 1–1.5 µg/kg, respectively. These values well-correlate with other papers published in the literature, such as that by Kosicki et al. (2016), who reported that 5% of maize samples were contaminated by AFs, with a mean value of 1.3 µg/kg [49]. Relative to the year 2015, the difference increased with the average of grain maize to 15.7 µg/kg, while that of high-moisture maize and silage maize was 4.8 and 2.2 µg/kg, respectively. Regarding silage maize, the average of AFB1 was relatively constant for all the years, except for a slight increase in 2015. 

Silage and high-moisture maize are used for feeding ruminants, while wet grain, preserved in the form of high-moisture maize, is widely used both in feeding ruminants and pigs. Maize grain flour is widely used for the feeding of all animal species and, in this form, represents the cereal most widely introduced in feed [28]. Maize by-products are included in both ruminant and monogastric diets, with their differences depending on the characteristics of the individual by-products. Maize is used for all livestock species and is a key ingredient that can hardly be replaced, given its high energy value and chemical composition.

The USDA estimates that global maize production was 1123.28 million tons in 2021. The main global producers were the USA, China, Brazil, the European Union, Argentina, Ukraine, India, Mexico, South Africa, Russia, and Canada. The main exporters, globally, were the USA, Argentina, Ukraine, Brazil, the European Union (France, Romania, and Hungary), Russia, South Africa, India, and Serbia. In 2020, 62.7 million tons of maize grain was produced in Europe; Italy was the fourth European producer behind Romania, with France and Hungary with a production of 6.8 million tons. Italy remains a strong importer, and in 2019–2020, almost 6 million tons of grain was imported, and the rate of self-supply was around 50%.

At the global level, there are some studies related to mycotoxin occurrence in grains, but they are scarce when referring to maize production and specifically for AFs [50]. 

Overall, the results confirm a year-to-year variation of AFB1 occurrence in maize, representing a problem of mycotoxin management. Aflatoxin production depends on environmental factors, such as temperature or humidity, and climate change forces new dynamics in those contaminants [36]. 

The conditions relating to flowering and then harvesting seem to be particularly delicate [24,28]. Aridity and bad agricultural practices are some of the causes that can lead to an increase in the AF level in crops. It is, therefore, important to study these increased levels, which depend on the matrix, and also to look at the variations that occur from year-to-year to discover the influence that the climate has on them [31]. This has been well-addressed also by Battilani et al. (2016) and by Gruber-Dorninger et al. (2019); Battilani and co-workers have highlighted how *Aspergillus flavus*, the key fungus for AF production, is well adapted to warm and dry weather conditions [31,51]. AFB1 is predicted to become a food safety issue in maize in Europe, especially in a warming scenario (+2 °C scenario), foreseen for the next years. These results represent a supporting tool to reinforce AF management and prevent human and animal exposure. Gruber-Dorninger et al. (2019) have reported that mycotoxin occurrence has a regional trend and, in several regions, mycotoxin concentrations in maize showed a pronounced year-to-year variation that could be explained by the rainfall or temperature during sensitive periods of grain development [28,31]. Thus, even though in the present study it was not possible to combine contamination levels with regional trends and climate patterns, an environmental effect, especially for maize (of local or EU origin), cannot be excluded. Indeed, according to Locatelli et al. (2022), 2015 and 2018 are the years in which the highest temperatures of the last 10 years were recorded in the Po Valley. In the study, the precipitation range and temperature were observed over the years, with values between 152.08–350.33 mm and 20.75–23.76 °C, respectively [28]. The year 2015 especially showed the harshest conditions, with high temperatures (23.46 °C), which were counterbalanced by low rainfall (155.62 mm) with respect to 2018 (209.83 mm) [28].

In 2003, the summer was particularly dry and hot, with maize crops stressed by drought and lack of water with the consequence of high contamination [34]. This was the first outbreak in Italy and alert in Europe related to the climate change effects on cultivated crops. A similar situation occurred in Italy during the summer of 2015 and until 2016, with an increase in AFB1 mean levels in the maize samples analysed. 

Climate change is responsible for increasing air temperature and CO_2_ concentration, and different rain distribution [51,52] and intensity, which sometimes generates extreme events with very high temperatures and an exceptional amount of rain, changing their frequency and variability. This situation seems in line with the scenario proposed by Battilani et al. (2016) for Europe, which suggested that AF contamination of maize kernels is an important concern for food and feed safety [51]. The authors, indeed, have suggested that the occurrence of Afs in food and feed is a “hot issue” in Europe. Such corners are mainly due to several changes in plant–pathogen interactions, which climate change can affect negatively. A higher AF risk of contamination has at least two different effects: at low levels, AF contamination could increase human and animal population chronic exposures to these mycotoxins; contaminated crops in general, and maize in particular, would reduce the availability of maize for both food and particularly feed uses, affecting, in turn, the feed–food supply, too. 

Climate change is also responsible for increasing the presence of pest species and diseases, such as *Diabrotica virgifera virgifera* and *Ostrinia nubilalis*. The attack of the larvae of maize pyralid (*Ostrinia nubilalis*) and other miners is not a direct cause of fungal development, but *A. flavus* grows faster in decay-damaged caryopses because it is more exposed to mycelium penetration, and plants subject to infestations or under stress can induce greater mycotoxin synthesis. The larvae of a *Diabrotica* (*Diabrotica virgifera virgifera*) damaged root system exposes plants to greater water and nutritional stress. Insect attacks induce a significant drop in yields and, therefore, a proportional increase in the concentration of AFs. 

Geographic information systems (GIS) and geostatistics have been applied to describe patterns of the population density and strain composition of AF contamination, *A. flavus* [42]. Climate models and pest distribution models may be the basis to predict the presence of fungi and the consequent production of Afs by planning the cultivation strategies and treatments to preserve crops’ quality and safety [53,54]. The quantitative relationships between the temperature and physiology of pest species can quantify how pest distributions may change in the future as the climate continues to change [53]. 

The interaction between Aw, temperature, and CO_2_ affects the expression of AF biosynthetic genes, showing a limited effect on AF growth but a significant impact on gene expression, significantly increasing AFB1 production, in vitro and on maize grains, and opening new research scenarios (epigenetic) [55]. Among the future prospects, there are also biomarkers for the identification of mycotoxin contamination and the unregulated mycotoxins related to AFs (Aflatoxicol and Sterigmatocystin).

An integrated approach, whereby AFs are controlled at all stages, from the field to the table, is required for risk reduction. Such an approach includes targeted plant breeding practices, the enhancement of host plant resistance, and biological control methods, coupled with post-harvest technologies, such as proper drying and storage of potentially affected crop products, as well as the development of appropriate alternative uses. Therefore, removing the sources of contamination, promoting better agricultural and storage techniques, ensuring adequate resources are available for testing and early diagnosis, enforcing strict food safety standards, informing and educating consumers and farmers, promoting better livestock feeding and management, and creating general awareness about personal protection are some of the ways in which authorities can help to control AFs. Physical, chemical, biological, and nanoparticle approaches are used for minimizing and managing AFs in food/feed crops [1,3,36,56]. Authors should discuss the results and how they can be interpreted from the perspective of previous studies and of the working hypotheses. The findings and their implications should be discussed in the broadest context possible. Future research directions may also be highlighted.

## 4. Conclusions

This study brings to light a significant volume of data on the contamination of a wide range of products used for the production of both animal feed and for direct human consumption. The AF content for most of the samples was in compliance with EU regulatory levels. Finally, the decline in positivity in recent years is linked to the creation of the Regional Surveillance Plan for AFs, which has highlighted the crises and offered solutions. Chain monitoring is essential to ensure safe production and consumer protection. The activation of the special plan has reduced risk to the consumer. 

Co-occurrence and climate change affect future prospects for predicting climate change effects on the production and presence of AFs. Supplier monitoring and control and the reduced buying of contaminated raw materials, as well as performing an analysis of each batch, help reduce AFs’ spread.

## 5. Materials and Methods

### 5.1. Chemicals and Reagents

All chemicals used were of analytical grade quality or better. Water and methanol (HPLC grade) were all obtained from Merck (Darmstadt, Germany). The quantitative filter paper, Whatman, was obtained from Thermo Fisher Scientific Inc. (Waltham, MA, USA). An Elisa kit BIO-SHIELD B1 5 was purchased from ProGnosis Biotech (Larissa, Greece).

### 5.2. Samples and Sample Preparation

For this study, a total of 10,280 samples, collected by the Regional Breeders Association of Lombardy between September 2013 and July 2021, were considered. Samples collected are reported in Table 2. Sampling was carried out on a wide category of matrices, namely: cotton, wheat, alfalfa, maize, soy, and total mixed rations. All matrices sampled and the number of samples for each matrix are specified in Table 1. 

The samples were collected from the Breeders of Lombardy and stored in plastic bags before analysis. Samples with moisture content higher than 15% were dried in a stove at 60 °C for 40 h and subsequently grounded with a knife mill with a 1 mm sieve (Retsch GmbH, Hann, Germany) before AFB1 analysis. For the extraction of AFB1, 20 g of the sample was weighed, and a solution of methanol/water, 70:30 *v*/*v* (100 mL), was added to the sample, and the suspension was placed under mechanical stirring for about 1 h. The solution was filtered on a Whatman paper filter and then analysed. In the case of the silage samples, a pH correction to a value of around 6 was made after filtration and before the analysis.

### 5.3. Enzyme-Linked Immunosorbent Assay (ELISA) Procedure

The Regional Breeders Association of Lombardy routinely analyses samples of different matrices through commercially available ELISA kits to constantly monitor the incidence of these contaminants. All samples were analysed using a competitive Enzyme-Linked Immunosorbent Assay (ELISA) BIO-SHIELD B1 5 test, purchased from ProGnosis Biotech, designed for the determination of AFB1 in grains, nuts, spices, cereals, and other commodities, including animal feed. This ELISA kit allows a limit of detection of 1.5 µg/kg, while quantification is possible between 2–50 µg/kg. Samples above the limit of quantification (50 µg/kg) were appropriately diluted and then retested. Duplicate analyses were performed, and the repeatability and recovery were observed using negative and positive controls, negative feed samples, and negative feed samples made positive with 20 µg/kg of AFB1.

### 5.4. Statistical Analysis

Results were analysed using MS Excel. Data are expressed as average contents, detection rates, and the percentage of samples that exceeded regulatory limits.

## Figures and Tables

**Figure 1 toxins-14-00763-f001:**
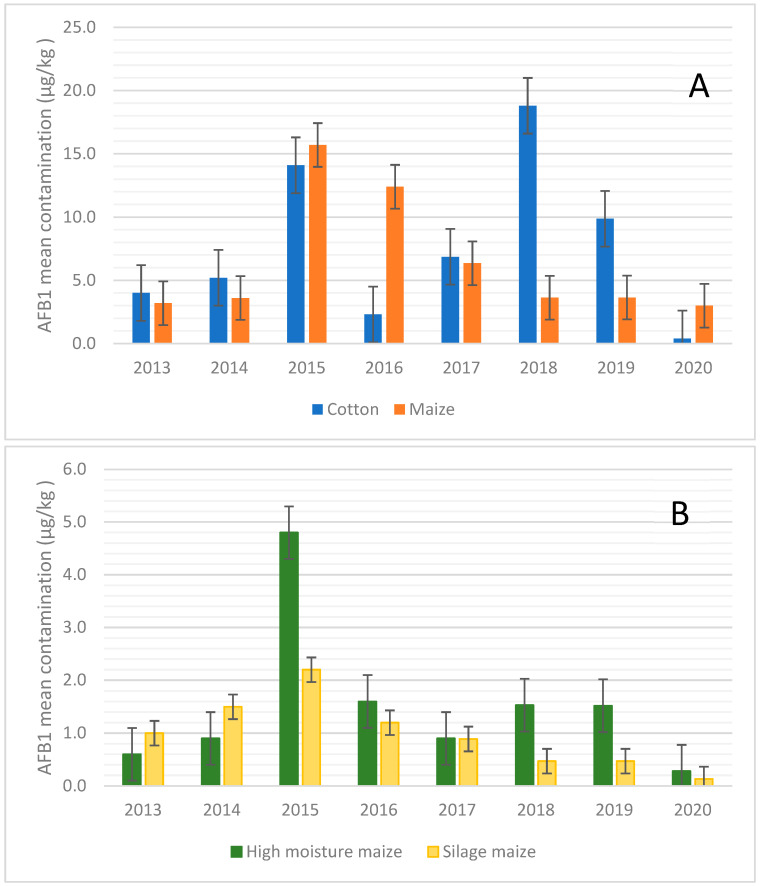
Aflatoxin B1 mean values over the years from 2013–2020. (**A**) Aflatoxin B1 mean values for cotton and maize samples over the years from 2013–2020; (**B**) Aflatoxin B1 mean values for high- moisture maize and silage maize samples over the years from 2013–2020.

**Figure 2 toxins-14-00763-f002:**
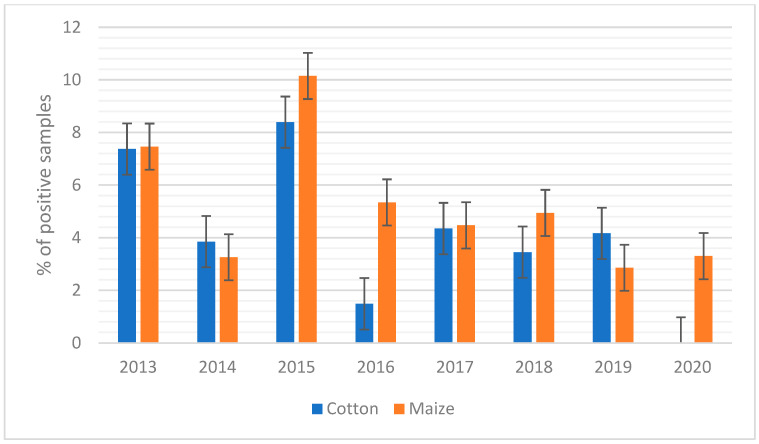
Aflatoxin B1 (AFB1) positive samples over the years from 2013–2020 (positive samples are defined as the number of samples that exceed the legal limit applied by the European Union (>20 µg/kg)). Percentage of positive samples of cotton (blue) and percentage of positive samples of maize (orange).

**Table 1 toxins-14-00763-t001:** Aflatoxin B1 occurrence in different feed materials.

Positive Samples							
	n ^1^	n ^2^	n ^2^ (%)	n ^3^	n ^3^ (%)	Mean Contamination (µg/kg)	Maximum Contamination (µg/kg)
Cotton	480	55	11.5	26	5.4	8.3	728.7
Wheat	52	0	0	0	0	<5	/
Maize (flour and grain)	5278	1256	23.8	534	10.1	9.1	1074
Alfalfa hay	19	1	5.3	0	0	<5	/
Silage maize	2003	64	3.2	6	0.3	<5	35.2
High-moisture maize	1364	122	8.9	19	1.4	<5	200
Soy	64	3	4.7	1	1.6	<5	>20
Unifeed	293	7	2.4	1	0.3	<5	>20

^1^ Number of samples analysed; ^2^ contaminated samples are defined as the number of samples that resulted positive for AFB1, even below the EU legal limit; ^3^ positive samples are defined as the number of samples that exceed the legal limit applied by the European Union (>20 µg/kg).

**Table 2 toxins-14-00763-t002:** Samples collected by the Regional Breeders Association of Lombardy for each type of feed material.

Cotton	Wheat	Maize	Alfalfa	HMM	SM	Unifeed	Soy
480	52	5278	19	1364	2003	293	64

Maize = Maize (flour and grains), HMM = High-Moisture Maize, SM = Silage Maize.

## Data Availability

Data not available to be shared.

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
