# Peer review of "An Eight-Year Survey on Aflatoxin B1 Indicates High Feed Safety in Animal Feed and Forages in Northern Italy"

_toxins, 2022, doi:10.3390/toxins14110763_

Round 1

Author Response

We would like to thank the reviewers for taking the time to read through our paper. We are also grateful to the reviewers for their comments.

Specifically:

Review 1

The present study entitled “An Eight-Year Survey on Aflatoxin B1 indicates high feed 2 safety in animal feed and forages in Northern Italy” provide very valuable information about the contamination of AFB1 in feeds. The number of samples included within the study alongside the chronological perspective allow to have a consistent overview of the impact of AFB1 in different typologies of feeds. Nevertheless, the manuscript would be much more relevant if results were better presented and discussed, among other drawbacks that must be address

Authors should indicate, at least in Table 1, the number or percentage of positive samples (intended as samples that tested positive for AFB1 even below the ML). This will be helpful to better understand the mean concentration values and the proportion of samples exceeding the ML out of all the positive samples. Additionally, the number of positive samples, even if compiling the current legislation, is also valuable toxicological information especially in a context of chronic exposure as authors mentioned in Lines 38-41.

Authors’ Reply: Thank you for your comments and suggestions. As a result, the number and also the percentage of positive samples, intended as positive results for AFB1, were included in Table 1.

Figure 2 must be rearranged; some information is missing due to formatting. Also, authors should consider integrating results into the same graphic for a better comparison between both matrices, just as Figure 1.

Authors’ Reply: Figure 2 has been changed according to your suggestions.

Lines 168-188: This whole part is just a descriptive section that do not provide any justification or confrontation to the obtained results in order to explain year-to-year variation, which is the title of this section. Therefore, I suggest to either place it into Introduction or remove it and re-elaborate this part of the Discussion.

Authors’ Reply: As suggested the whole part has been re-elaborated to make it less descriptive (line 180-203).

Line 240 and on: Why authors put together all this information within in the “Maize, maize silage and high moisture maize” section? This can stand as a single section inside Discussion or either the information should be redistributed among the sections established for each matrix.

Authors’ Reply: Thank you for your comment. Information has been redistributed in the sections of each matrix (line 188-203 and 255-258)

Could authors provide more information about the method performance? Repeatability and reproducibility are important parameters when a methodology is applied over long periods of time, as in the case of the present study.

Authors’ Reply: We are grateful, thanks for underlying this point. Repeatability and reproducibility are indeed important parameters. For this reason, we have added a description in the text (line 380-381) of how they are monitored over the long-term periods.

Line 368: authors stated that with this ELISA kit, “quantification is possible between 2-50µg/kg”. Then why the maximum contamination in cotton and maize exceeded this value? Authors need to clarify this.

Authors’ Reply: Thanks for underling this point. A description on how quantification was performed has been implemented (lines 389-379).

Minor comments:

Line 94 and on: All aflatoxins abbreviations have been properly detailed within the introduction section. Once explained, authors should use the abbreviations and be consistent throughout the whole text.

Authors’ Reply: Done.

Line 146: Limit of detection is said to be 5 µg/kg whereas Line 367 reflects 1.5 µg/kg. Please clarify this.

Authors’ Reply: Thanks for your suggestion. Limit of detection is 1.5µg/kg, while limit of quantification is 2µg/kg. We have been clarified it in lines 377-378.

Lines 157-158: Authors should describe the statistical analysis and results in order to confirm a significant year-to-year variation in the occurrence of AFB1 in cotton and maize.

Authors’ Reply:

There are some English typos such as:

Line 148: Change “This” to “These”.

Authors’ Reply: Done.

Line 149: Change “reached” to “reach”

Authors’ Reply: Done.

Please check the English

Authors’ Reply: English has been revised.

Reviewer 2 Report

The authors collected data from n eight years survey on aflatoxin contamination. The aim was to present a significant volume data in Northern Italian feed ingredients.

In the introduction Latin taxonomy you should use on first appearance of a plant (i.e. lane 54) or not at all.

In lane 78-82, in average numbers with lots of zeros are disturbing. Make it easier and comparable e.g. in ng/kg.

In Figure 1: there is no sign for a or b figure, there is no standard deviation on the bars. I could not agree to use ug/kg and ppb also on the figures. The authors use SI in the article, why we would use ppb here?

In Figure 2 also there is no sign for a or b figure, there is no standard deviation on the bars, and the a and b figure are covering each other.

In Discussion

Lane 133 change aflatoxins for aflatoxin, and at several other case.

Lane 149 cannot reach the market

Lane 154 the higher aflatoxin contamination of maize and not wheat is a well-known characteristic. Check lipid’s role in aflatoxin regulation.

Lane 174 I do not understand: (Utilizable energy content of the feedstuffs). the percentages are energy contents??

For 3.2.1. last paragraph you need references

Around lane 253-269, I think this part is better in Introduction.

I do not see the article round. The aim is shallow, we needed more important aims, than to show off big numbers. I think the discussion, there is a missing message. The message is that everything is ok?

What about the climate data on Northern Italy. How does it support aflatoxin content? Are there a correlation?

Author Response

We would like to thank the reviewers for taking the time to read through our paper. We are also grateful to the reviewers for their comments.

Specifically:

Review 2

The authors collected data from n eight years survey on aflatoxin contamination. The aim was to present a significant volume data in Northern Italian feed ingredients.

In the introduction Latin taxonomy you should use on first appearance of a plant (i.e. lane 54) or not at all.

Authors’ Reply: Latin name has been eliminated.

In lane 78-82, in average numbers with lots of zeros are disturbing. Make it easier and comparable e.g. in ng/kg.

Authors’ Reply: Done.

In Figure 1: there is no sign for a or b figure, there is no standard deviation on the bars. I could not agree to use ug/kg and ppb also on the figures. The authors use SI in the article, why we would use ppb here?

Authors’ Reply: Thank you for your suggestions. Figure 1 has been revised according to the comments.

In Figure 2 also there is no sign for a or b figure, there is no standard deviation on the bars, and the a and b figure are covering each other.

Authors’ Reply: Figure 2 has been revised according to your comments.

In Discussion

Lane 133 change aflatoxins for aflatoxin, and at several other case.

Authors’ Reply: Done.

Lane 149 cannot reach the market

Authors’ Reply: Typo has been modified.

Lane 154 the higher aflatoxin contamination of maize and not wheat is a well-known characteristic. Check lipid’s role in aflatoxin regulation.

Authors’ Reply: We are grateful for emphasizing this point. We included these new evidences in the text (lines 163-166).

Lane 174 I do not understand: (Utilizable energy content of the feedstuffs). the percentages are energy contents??

Authors’ Reply: The paragraph has been modified eliminating understandable concepts (lines 180-183).

For 3.2.1. last paragraph you need references

Authors’ Reply: The paragraph has been modified including appropriate references (lines 188-203).

Around lane 253-269, I think this part is better in Introduction.

Authors’ Reply: According to your suggestion, the interested part has been moved in the Introduction (lines 56-63).

I do not see the article round. The aim is shallow, we needed more important aims, than to show off big numbers. I think the discussion, there is a missing message. The message is that everything is ok? Authors’ Reply: Based on our knowledge, in all the various surveys on the presence of mycotoxins, one of the conclusions is that there are samples that exceed legal limits also in relation to the different years analysed, climatic-environmental conditions, etc., although most of the samples comply with regulatory limits. For this reason, authors think continuous monitoring is fundamental. "Big numbers" is definitely a focal point of this study, underlining how the continuous monitoring represents the first step for the safety of the feed chain.

What about the climate data on Northern Italy. How does it support aflatoxin content? Are there a correlation?

Authors’ Reply: In this article, authors only monitored aflatoxins contamination without recording the climatic conditions of the affected period. However, a correlation between climatic conditions and aflatoxins contamination can be found by looking at some data published in the literature in the same period in the Po valley. This correlation is well explained in the text (line 276-281) with the quote of the appropriate article.

Round 2

Reviewer 2 Report

I am glad about the improvements; however, besides some spelling problems, I miss the statistical analysis of the data. Why can we not see the significant changes over the years proven by statistical analysis? In methods, also you should put your statistical analysis method.

Lane 59: enhancing

Lane 63. needs a Ref.

Lane 320 Sterigmatocystin

Author Response

Dear REviwer thansk for you valuable comment: . Why can we not see the significant changes over the years proven by statistical analysis? In methods, also you should put your statistical analysis method. 

AU: In this study, the authors opted for a descriptive statistical analysis of the data rather than an inferential analysis. The data are in fact the result of a monitoring campaign in feed in northern Italy and the purpose of this study was precisely to highlight how monitoring campaigns like this help to keep AFB1 contamination within the legal safety limits. The focus was therefore an analysis that showed the importance of control and monitoring along the supply chain and not the significance of the differences in AFB1 contamination between different years. Moreover, with samples resulting from a monitoring campaign like this, a statistical approach different from the descriptive one could lead to misleading conclusions given the great variability of the matrices involved and an unbalance number of samples between different matrices.

Lane 59: enhancing 

AU: corrected 

Lane 63. needs a Ref.

AU: added

Lane 320 Sterigmatocystin

AU: corrected 

Regards